

# Differential effects of cotreatment of the antibiotic rifampin with host-directed therapeutics in reducing intracellular *Staphylococcus aureus* infection

Melissa D. Evans[1], Robert Sammelson[2] and Susan McDowell[1]

[1] Department of Biology, Ball State University, Muncie, IN, United States of America
[2] Department of Chemistry, Ball State University, Muncie, IN, United States of America

Corresponding author
Melissa D. Evans,
melissa.evans96@outlook.com

## ABSTRACT

**Background**. Chronic infection by *Staphylococcus aureus* drives pathogenesis in important clinical settings, such as recurrent pulmonary infection in cystic fibrosis and relapsing infection in osteomyelitis. Treatment options for intracellular *S. aureus* infection are limited. Rifampin, a lipophilic antibiotic, readily penetrates host cell membranes, yet monotherapy is associated with rapid antibiotic resistance and development of severe adverse events. Antibiotic cotreatment can reduce this progression, yet efficacy diminishes as antibiotic resistance develops. ML141 and simvastatin inhibit *S. aureus* invasion through host-directed rather than bactericidal mechanisms.

**Objective**. To determine whether cotreatment of ML141 or of simvastatin with rifampin would enhance rifampin efficacy.

**Methods**. Assays to assess host cell invasion, host cell viability, host cell membrane permeability, and bactericidal activity were performed using the human embryonic kidney (HEK) 293-A cell line infected with *S. aureus* (29213) and treated with vehicle control, simvastatin, ML141, rifampin, or cotreatment of simvastatin or ML141 with rifampin.

**Results**. We found cotreatment of ML141 with rifampin reduced intracellular infection nearly 85% when compared to the no treatment control. This decrease more than doubled the average 40% reduction in response to rifampin monotherapy. In contrast, cotreatment of simvastatin with rifampin failed to improve rifampin efficacy. Also, in contrast to ML141, simvastatin increased propidium iodide (PI) positive cells, from an average of 10% in control HEK 293-A cells to nearly 20% in simvastatin-treated cells, indicating an increase in host cell membrane permeability. The simvastatin-induced increase was reversed to control levels by cotreatment of simvastatin with rifampin.

**Conclusion**. Taken together, rifampin efficacy is increased through host-directed inhibition of *S. aureus* invasion by ML141, while efficacy is not increased by simvastatin. Considerations regarding novel therapeutic approaches may be dependent on underlying differences in pharmacology.

## INTRODUCTION

*Staphylococcus aureus* is an important cause of both acute (*Tong et al., 2015*) and chronic infections (*Kavanagh et al., 2018*; *Goss & Muhlebach, 2011*). Chronic infections contribute substantially to morbidity and mortality in certain settings, most notably in progressive lung disease characteristic of cystic fibrosis (CF) (*Goss & Muhlebach, 2011*) and in deterioration of bone and joint tissue in osteomyelitis (*Kavanagh et al., 2018*). *S. aureus* is an initial pulmonary isolate in pediatric patients with CF (*Armstrong et al., 1997*) and by adulthood the majority of CF patients remain chronically infected (*Branger, Gardye & Lambert-Zechovsky, 1996*; *Schwerdt et al., 2018*). Chronic *S. aureus* infection is an ongoing treatment challenge as indicated by the 2018 Cystic Fibrosis Foundation Patient Registry reporting an increase in the percentage of patients infected with *S. aureus* each year from 59% in 2003 to 70% in 2018 (*Cystic Fibrosis Foundation, 2019*). *S. aureus* also is the most common cause of acute and chronic osteomyelitis in children and adults (*Kavanagh et al., 2018*; *Hatzenbuehler & Pulling, 2011*). Treatment of staphylococcal osteomyelitis is complicated further by increased incidence of methicillin-resistant *S. aureus* (MRSA) infection. The predominance of *S. aureus* in initiating and sustaining chronic infection may be attributable to the capacity to invade host cells, to reemerge and invade adjoining cells and to undergo phenotypic differentiation within host cells enabling persistence within the intracellular environment (*Loffler et al., 2014*; *Proctor et al., 2006*; *Tuchscherr et al., 2011*).

*S. aureus* invades host cells by exploiting host endocytic mechanisms (*Foster et al., 2014*). On the bacterial surface, invasive *S. aureus* strains express fibronectin binding proteins (FnBPs) that bind fibronectin, a host extracellular matrix protein. Fibronectin, as it binds to the host cell receptor $\alpha 5 \beta 1$, triggers receptor-mediated endocytosis of the bacteria-fibronectin complex. During invasion, host cell actin stress fibers disassemble, potentially providing pulling forces needed for engulfment (*Horn et al., 2008*; *Agerer et al., 2005*). CDC42, a member of the RHO GTPase family, regulates actin stress fiber dynamics and can function ahead of family members RAC and RHO in the mobilization of actin (*Nobes & Hall, 1995*). The apparent regulatory role and early CDC42 activation during *S. aureus* invasion (*Arbibe et al., 2000*) suggest host CDC42 plays a central role in this invasive mechanism.

Treatment options for intracellular *S. aureus* infection are limited as first-line antibiotics demonstrate limited membrane permeability (*Darouiche & Hamill, 1994*). Rifampin is a lipophilic antibiotic that demonstrates a propensity for intracellular uptake, enabling clearance of both extracellular and intracellular susceptible bacteria (*Hoger et al., 1985*; *Mandell & Vest, 1972*). However, rifampin monotherapy is associated with the development of cross resistance to vancomycin and daptomycin (*Guerillot et al., 2018*), rapid rifampin resistance (*Zhou et al., 2012*; *Bongiorno et al., 2018*), and progression of severe adverse events (*Poole, Stradling & Worlledge, 1971*; *Grosset & Leventis, 1983*). To limit the development of resistance and adverse events, the current standard of care is cotreatment of rifampin with an antibiotic cocktail (*Forrest & Tamura, 2010*).

To circumvent antibiotic resistance, an emerging therapeutic approach is to target the host rather than bacterial cells (*Horn et al., 2008*; *Cordero et al., 2014*). For nearly two

decades, researchers have examined statin drugs as potential host-directed therapeutics for infection, including invasive infection by *S. aureus* (*Zumla et al., 2016*; *Hennessy et al., 2016*; *Caffrey et al., 2017*; *Parihar, Guler & Brombacher, 2019*; *Kaufmann et al., 2018*). Our work had indicated therapeutic benefit may include limiting the spread of infection by reducing host cell invasion (*Burns et al., 2013*; *Smelser et al., 2016*; *McDowell et al., 2011*). We identified an underlying mechanism where simvastatin reduces invasion of *S. aureus* into host cells by sequestering small-GTPases CDC42, RAC, and RHO in the cytosol (*Horn et al., 2008*). In turn, this sequestration limits actin stress fiber disassembly and reduces fibronectin binding at $\alpha5\beta1$ (*Caffo et al., 2019*). We went on to investigate whether targeted inhibition of host CDC42 would be sufficient to limit invasion. We discovered that ML141, a small molecule inhibitor with specificity for host CDC42 (*Hong et al., 2013*; *Surviladze et al., 2010*), decreased host cell invasion (*Cordero et al., 2014*). Similar to simvastatin, we found ML141 inhibition of invasion is associated with diminished reordering of actin stress fibers and with decreased fibronectin binding at the host cell membrane. Given simvastatin and ML141 reduce the number of bacteria that invade the host cell and rifampin acts on both extracellular and intracellular bacterial populations, we sought to determine in the current study whether cotreatment of these host-directed inhibitors with rifampin would enhance clearance of intracellular infection. We examined the response to cotreatment on intracellular infection, bactericidal activity, host cell membrane permeability, and host cell viability.

## MATERIALS & METHODS

### Human Embryonic Kidney (HEK) 293-A cell culture

HEK 293-A cells (Fisher Scientific) were cultured in Dulbecco's Modified Eagle's Medium (VWR International, Radnor, PA; DMEM) supplemented with 10% fetal bovine serum (Atlanta Biologicals, Flowery Branch, GA; FBS) and 1% L-glutamine (Fisher Scientific) and maintained at 37 °C, 5% $CO_2$, in 75 cm$^2$ vented cap flasks (Fisher Scientific). HEK 293-A cell identity was authenticated by short tandem repeat profiling performed by American Type Culture Collection (ATCC, Manassas, VA).

### Bacterial cell culture

Two days prior to each assay, *Staphylococcus aureus* (ATCC, #29213) cultures were maintained in 5 ml tryptic soy broth (Sigma-Aldrich, St. Louis, MO) overnight (37 °C, 225 rpm) then subcultured one day prior to each assay.

### Rifampin IC50

*S. aureus* cultures were harvested by centrifugation (3 min, 37 °C, 10,000 rpm), resuspended to $5.4 \times 10^8$ colony forming units (CFU)/ml in prewarmed 0.85% saline, and incubated with the vehicle control, 0.4% dimethyl sulfoxide (Fisher Scientific, Waltham, MA; DMSO), or with increasing concentrations of rifampin (VWR International; structure provided in Supplemental File) (1 hr, 37 °C, 5% $CO_2$). Serial dilutions were incubated overnight on tryptic soy agar (Sigma-Aldrich; TSA; 37 °C) and colony counts performed to determine CFU/ml. Of note, for all experiments, 0.4% was the final DMSO solvent concentration

(vol/vol) for each treatment and 0.4% DMSO served as the non-treatment, negative, vehicle control.

To determine the IC50 of rifampin for intracellular infection, 35 mm cell culture dishes (Fisher Scientific) were precoated with Attachment Factor (Fisher Scientific) prior to plating $3 \times 10^5$ HEK 293-A cells/ml. HEK 293-A cells were inoculated with *S. aureus* at multiplicity of infection (MOI) 2 ($6 \times 10^5$ CFU) or at MOI 100 ($3 \times 10^7$ CFU; 30 min or 1 hr, 37 °C, 5% $CO_2$) in 10% FBS/phosphate buffered saline (VWR International; PBS) followed by incubation with increasing concentrations of rifampin (1 hr, 37 °C, 5% $CO_2$). To remove extracellular bacteria, HEK 293-A cells were incubated with gentamicin (Sigma-Aldrich; 50 µg/ml) and lysostaphin (Sigma-Aldrich, 20 µg/ml) in DMEM (45 min, 37 °C, 5% $CO_2$) following three 1X PBS washes. Intracellular *S. aureus* were harvested using 1% saponin (20 min, 37 °C, 5% $CO_2$) following three 1X PBS washes and serial dilutions of the supernatant were plated onto TSA, incubated overnight at 37 °C and CFU/ml quantified by colony counts.

### ML141 pretreatment assay

HEK 293-A cells were plated as described above. The following day, HEK 293-A cells were pretreated with DMSO (0.4%) or with ML141 (10 µM; 37 °C, 5% $CO_2$, 24 h; structure provided in Supplemental File). The next day, pretreated HEK 293-A cells were inoculated ($3 \times 10^7$ CFU, 1 hr, 37 °C, 5% $CO_2$) in 10% FBS/PBS. Intracellular bacteria were isolated and quantified as described above.

### Cotreatment assay

HEK 293-A cells were plated as described above. Treatments were performed as outlined in Table 1 at these concentrations: DMSO (0.4%), ML141 (10 µM), simvastatin (1 µM; structure provided in Supplemental File), rifampin (0.01 mg/L).

Following post-treatment, intracellular bacteria were isolated and quantified as described.

### Host cell viability assay

HEK 293-A cells were plated at $5 \times 10^4$ cells/0.5 ml in 48-well cell culture plates (VWR International) precoated with Attachment Factor. To acquire uniform microplate readings, multi-well culture plates were used. This change from the 35 mm plates used for the invasion assay required a reduction in cell count. The following day, HEK 293-A cells were treated with DMSO (0.4%), ML141 (10 µM), rifampin (0.01 mg/L), or ML141 (10 µM) with rifampin (0.01 mg/L; 24 hr, 37 °C, 5% $CO_2$). Cell viability was measured using CellTiter 96® Aqueous One Solution Reagent (Promega Corporation, Madison, WI). Absorbance was measured at 490 nm using a BioRad iMark microplate reader.

### Bactericidal assay

*S. aureus* were harvested as described above and treated with DMSO (0.2%), ML141 (10 µM), rifampin (0.002 mg/L), or ML141 (10 µM) with rifampin (0.002 mg/L; 1 h, 37 °C, 5% $CO_2$). Bacteria were serially diluted, plated on TSA, incubated overnight at 37 °C and CFU/ml determined from colony counts.

**Table 1  Treatment regimen for cotreatment assays.**

| | Treatment Group | | | |
|---|---|---|---|---|
| | DMSO | ML141 or simvastatin | Rifampin | ML141/rifampin or simvastatin/rifampin |
| Pre-treatment(24 hr) | DMSO | ML141 or simvastatin | DMSO | ML141 or simvastatin |
| Inoculation(30 min) | $3 \times 10^7$ CFU | $3 \times 10^7$ CFU | $3 \times 10^7$ CFU | $3 \times 10^7$ CFU |
| Post-treatment(1 hr) | DMSO | ML141 or simvastatin | Rifampin | ML141/rifampin or simvastatin/rifampin |

### Propidium iodide flow cytometry assay

HEK 293-A cells were plated and incubated with DMSO (0.4%), ML141 (10 μM), or simvastatin (1 μM) as described in the cotreatment assay. The following day, HEK 293-A were treated with DMSO (0.4%), ML141 (10 μM), simvastatin (1 μM), rifampin (0.01 mg/L), ML141 (10 μM) with rifampin (0.01 mg/L), or simvastatin (1 μM) with rifampin (0.01 mg/L; 1 hr, 37 °C, 5% $CO_2$). Immediately prior to performing the assay, HEK 293-A cells serving as the positive control were incubated with 70% ethanol (1 min). HEK 293-A cells were harvested by scrapping with cell lifters and washed in FACS buffer. Propidium iodide (Sigma Aldrich) was added to each sample immediately prior to measurement. Percentage of propidium iodide positive HEK 293-A cells was determined using a MACSQuant Analyzer 10 flow cytometer (Miltenyi Biotech Inc., Auburn, CA.).

### Statistical analysis

All data were analyzed using Prism software (GraphPad Software Inc., San Diego, CA). Rifampin IC50 was determined by nonlinear regression analysis. Means between groups were compared by one-way ANOVA followed by Newman-Keuls or Holm-Sidak's post-hoc analysis. The threshold for statistical significance was $P < 0.05$.

## RESULTS

### Cotreatment of rifampin with ML141 reduces intracellular *S. aureus* infection more than rifampin alone

We first determined the half maximal inhibitory concentration (IC50) of rifampin in a model cell line, HEK 293-A. The rifampin IC50 was 0.004 mg/L when HEK 293-A cells were incubated with bacteria at MOI 2 and the IC50 increased to 0.01 mg/L when the MOI was increased to 100 (Fig. 1). Thus, a higher concentration of antibiotic was needed to clear a larger intracellular bacterial population. All subsequent experiments were performed at MOI 100 to ensure there were sufficient numbers of colonies in the cotreatment group for statistical analysis.

Consistent with what we had found in human umbilical vein endothelial cells (*Cordero et al., 2014*), pretreatment of HEK 293-A cells with 10 μM ML141 reduced the number of intracellular bacteria by an average of 40% when compared to pretreatment with the negative, vehicle control DMSO (Fig. 2A, $P = 0.0169$). Therefore, this concentration of ML141 served as the positive control for all subsequent experiments. Also consistent with our work and that of others (*Chow et al., 2010*; *Merx et al., 2004*), the experimental design used throughout this study is one of pretreatment. Pretreatment is not used as a model for

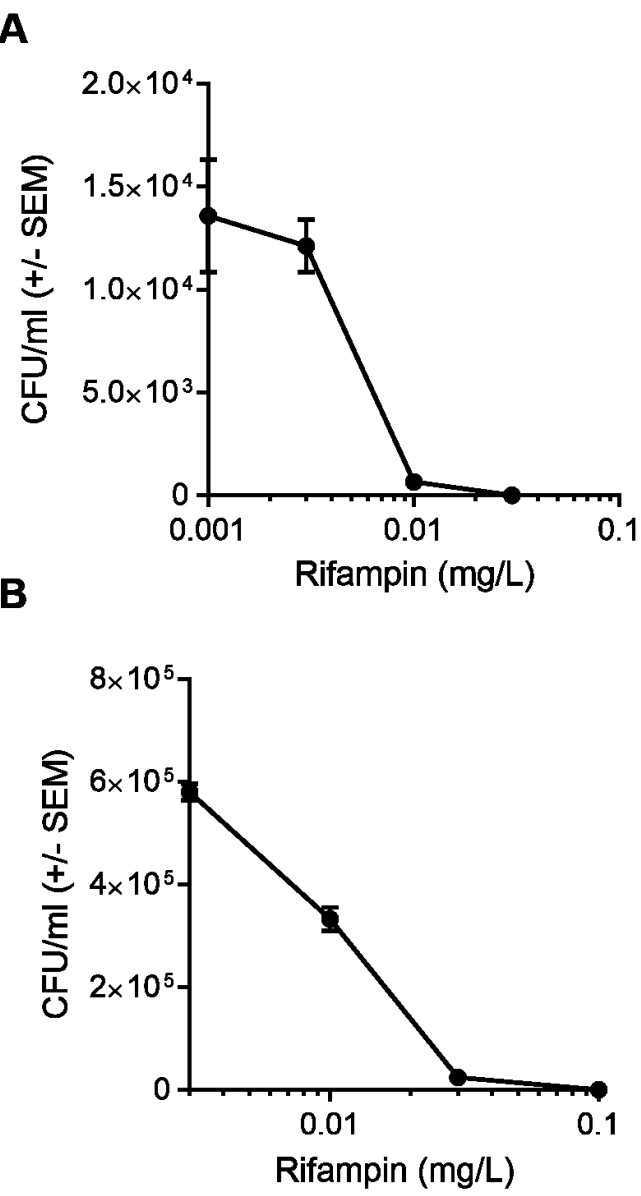

**Figure 1  A higher concentration of rifampin is required when there is an increase in the bacterial load.** (A) HEK 293-A cells were plated at $3 \times 10^5$ cells/ml in 35 mm cell culture dishes. Two days later, cells were infected at a multiplicity of infection (MOI) of 2 ($6 \times 10^5$ *Staphylococcus aureus* colony forming units; CFU; 30 min, 37 °C, 5% $CO_2$) then treated with dimethyl sulfoxide (DMSO), or 0.001, 0.003, 0.01, or 0.03 mg/L rifampin (1 hr, 37 °C, 5% $CO_2$). HEK 293-A cells were treated with gentamicin (50 $\mu$g/ml) and lysostaphin (20 $\mu$g/ml; 45 min, 37 °C, 5% $CO_2$) to remove extracellular bacteria and intracellular bacteria were harvested using 1% saponin (20 min, 37 °C, 5% $CO_2$). Serial dilutions were plated on tryptic soy agar (24 hr, 37 °C) and colony counts were performed. Data are pooled from two independent experiments and are represented as CFU/ml ± SEM ($n = 4 - 8$/treatment). (B) HEK 293-A cells were plated as described above and infected at MOI of 100 ($3 \times 10^7$ CFU; 1 hr, 37 °C, 5% $CO_2$). HEK 293-A cells were treated with DMSO, or 0.003, 0.01, 0.03, or 0.1 mg/L rifampin (1 hr, 37 °C, 5% $CO_2$). Extracellular bacteria were killed, intracellular bacteria harvested, and CFU/ml determined as described above. Data are represented as CFU/ml ± SEM ($n = 4$/treatment).

prophylactic clinical applications, but rather, as a model for limiting the spread of infection post-onset and diagnosis, given the evidence that invasive *S. aureus* strains successfully exit infected host cells to initiate new infection (*Tuchscherr et al., 2011*; *Gresham et al., 2000*; *Jubrail et al., 2016*). We found cotreatment of 10 μM ML141 with rifampin at the IC50 decreased the number of intracellular bacteria more than ML141 or rifampin alone (Fig. 2B, *P* < 0.0001). Thus, cotreatment of ML141 with rifampin appeared to enhance bacterial clearance.

## Underlying mechanisms for enhanced clearance are not associated with an appreciable loss in host cell viability or loss of host cell membrane integrity

We next assessed whether the reduction in the number of intracellular bacteria was due to decreased numbers of viable host cells rather than to enhanced bacterial clearance. HEK 293-A cells were incubated with the DMSO control, ML141 alone, rifampin alone, or ML141 combined with rifampin for the same length of time as had been used for the invasion assay. HEK 293-A metabolic activity was assayed by measuring the conversion of a tetrazolium compound to formazan, a colored product detectable by absorbance at 490 nm. Compared to the DMSO control group, no decrease in absorbance was detected in any treatment group, indicating treatment did not diminish metabolic activity, an indicator of sustained cell viability (Fig. 3A).

We went on to assess whether clearance of intracellular infection might be attributable to increased host membrane permeability, allowing greater penetrance of antibiotic into the intracellular compartment. HEK 293-A cells were incubated with the vehicle control DMSO or with ML141, rifampin, or ML141 combined with rifampin for the same duration as had been used for the invasion assay. Host cell membrane permeability was assessed using propidium iodide uptake. The percentage of propidium iodide positive HEK 293-A cells, an indication of membrane permeability, remained at DMSO control levels in response to each treatment, indicating HEK 293-A host cell membrane integrity is maintained following incubation with all treatments (Figs. 3B and 3C).

## Underlying mechanisms for enhanced clearance by the co-treatment are not associated with appreciable ML141 bactericidal activity or with enhanced rifampin bactericidal activity

Although ML141 has been assessed repeatedly for bactericidal activity and no such activity has been detected toward multiple *S. aureus* or *Streptococcus pyogenes* strains (*Cordero et al., 2014*; *Caffo et al., 2019*), we saw it as important to verify this under the experimental conditions of the rifampin studies. We inoculated HEK 293-A media containing the DMSO control or 10 μM ML141 for the same length of time used in the invasion assay. We found that the number of viable bacteria following ML141 treatment was similar to the control treated group, indicating ML141 exhibited no bactericidal activity under the conditions used for the invasion assay (Fig. 4B).

To assess whether co-treatment with ML141 might somehow enhance rifampin bactericidal activity, we first needed to determine the IC50 of rifampin on *S. aureus* ATCC 29213 growth in axenic (host cell-free) conditions. This was necessary because the

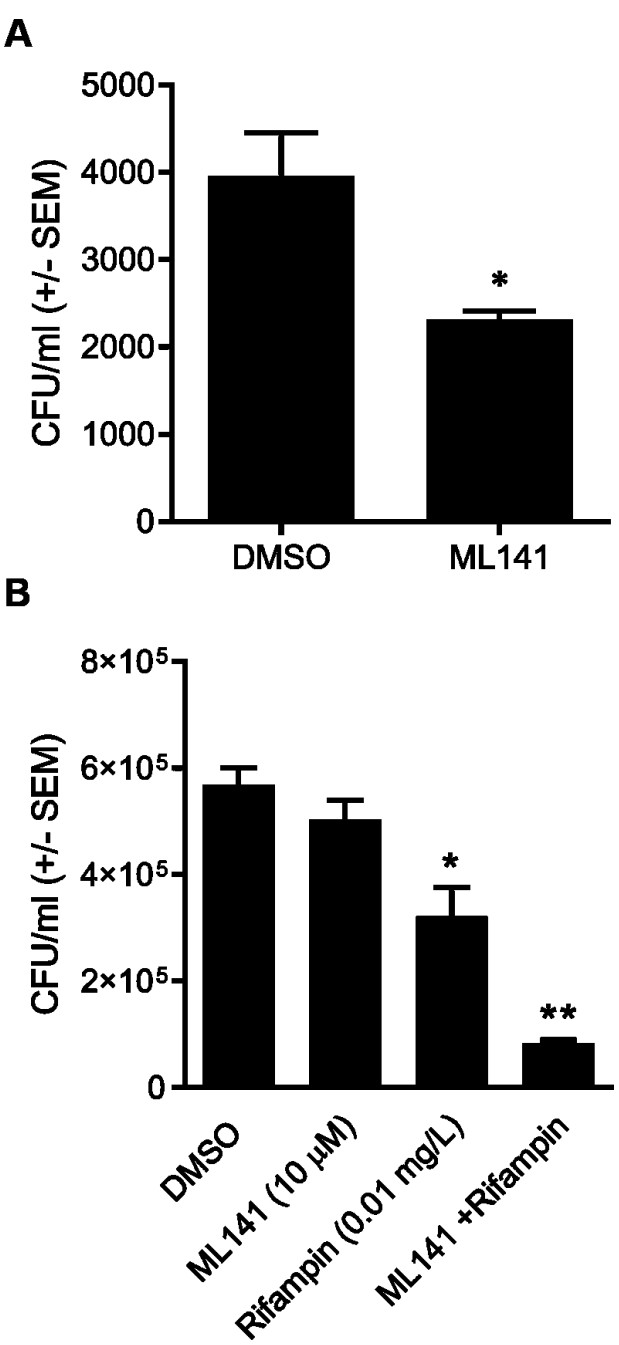

**Figure 2** **Pretreatment with ML141 and cotreatment of ML141 with rifampin reduces intracellular** *Staphylococcus aureus* **infection.** (A) HEK 293-A cells were plated at $3 \times 10^5$ cells/ml in 35 mm cell culture dishes. The following day, cells were treated with dimethyl sulfoxide (DMSO) or ML141 (10 μM; 24 hr, 37 °C, 5% $CO_2$). HEK 293-A cells were infected at a multiplicity of infection (MOI) of 2 (30 min, 37 °C, 5% $CO_2$) then were treated with gentamicin (50 μg/ml) 

**Figure 2 (…continued)**
and lysostaphin (20 µg/ml; 45 min, 37 °C, 5% $CO_2$) to remove extracellular bacteria. Intracellular bacteria were harvested using 1% saponin (20 min, 37 °C, 5% $CO_2$) and serial dilutions were plated on tryptic soy agar (24 hr, 37 °C). Intracellular bacteria were quantified by colony counts. Data are represented as colony forming units (CFU)/ml $\pm$ SEM (* less than DMSO, $P < 0.05$ by Student's $t$-test, $n = 4$/treatment). (B) HEK 293-A cells were plated and pretreated with ML141 or DMSO as described above. HEK 293-A cells were infected at MOI of 100 by incubating with $3 \times 10^7$ S. aureus CFU then were treated with DMSO, ML141 (10 µM), rifampin (0.01 mg/L), or cotreatment of ML141 (10 µM) with rifampin (0.01 mg/L; 1 hr, 37 °C, 5% $CO_2$). Extracellular bacteria were killed, and intracellular bacteria were harvested and plated as described above. Data are represented by CFU/ml $\pm$ SEM (* less than DMSO and ML141, ** less than DMSO, ML141, or rifampin, $P < 0.05$ by one-way ANOVA followed by Holm-Sidak's post-hoc analysis, $n = 4$/treatment). Data are representative of three replicate experiments.

rifampin IC50 for intracellular bacteria (0.01 mg/L, reported above) was expected to exceed the IC50 needed under axenic conditions. Consistent with this expectation, the IC50 of rifampin in the absence of host cells was 0.002 mg/L (Fig. 4A). We then found the number of viable bacteria following cotreatment of 10 µM ML141 with rifampin at the IC50 for axenic conditions was similar to the number of bacteria recovered following rifampin alone, indicating no detectable enhancement of rifampin bactericidal activity by ML141 at the concentrations used (Fig. 4C, $P = 0.0005$).

## Cotreatment of simvastatin with rifampin fails to improve rifampin efficacy

We went on to explore the hypothesis that cotreatment of rifampin with the host-directed therapeutic simvastatin might achieve similar enhancement in the clearance of intracellular infection. The hypothesis was based on our earlier findings that 1.0 µM simvastatin decreases intracellular infection in part by sequestering host GTPases, including CDC42 (*Horn et al., 2008*), the host-directed target of ML141 (*Hong et al., 2013*; *Surviladze et al., 2010*). Moreover, both ML141 and simvastatin disrupt actin dynamics necessary for host cell invasion by *S. aureus* and by *Streptococcus pyogenes* (*Cordero et al., 2014*; *Caffo et al., 2019*). Contrary to the hypothesis, we found no enhancement of bacterial clearance was achieved by cotreatment of rifampin with simvastatin (Fig. 5A, $P = 0.0124$).

## Differential effect of simvastatin on HEK 293-A cell membrane permeability is reversed by cotreatment

We were curious to understand the differential effect between simvastatin and ML141. In earlier work, we (*Caffo et al., 2019*) and others (*Chow et al., 2010*) had found that simvastatin can induce host cell membrane permeability in specific cell types. We examined the effect of simvastatin on HEK 293-A cell membrane permeability and the effect of rifampin cotreatment. We found in contrast to ML141 (Fig. 3B), simvastatin treatment increased HEK 293-A cell permeability (Fig. 5B, $P = 0.0042$). We also found the increase in membrane permeability in response to simvastatin was reversed by cotreatment with rifampin (Figs. 5B and 5C).

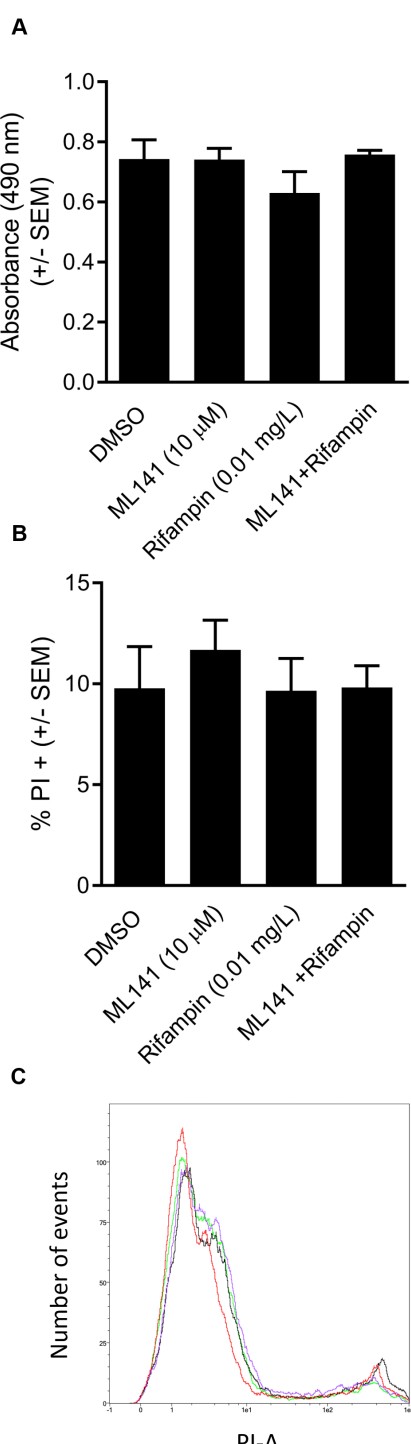

**A**

**B**

**C**

**Figure 3** **HEK 293-A cell viability and membrane integrity are maintained following cotreatment of ML141 with rifampin.** (A) HEK 293-A cells were plated at $5 \times 10^4$ cells/well in a 48-well plate. The following day, cells were treated with dimethyl sulfoxide (DMSO), ML141 (10 μM), rifampin (0.01 mg/L), or cotreatment of ML141 (10 μM) and rifampin (continued on next page…)

**Figure 3 (…continued)**
(0.01 mg/L; 24 hr, 37 °C, 5% $CO_2$). Host cell viability was quantified following incubation with the CellTiter 96® AQueous One Solution Reagent (1 hr, 37 °C, 5% $CO_2$). Absorbance was measured at 490 nm using a BioRad iMark Microplate Reader. Data are represented as absorbance ± SEM ($P > 0.05$ by one-way ANOVA followed by Holm-Sidak's post-hoc analysis, $n = 4$/treatment). (B) HEK 293-A cells were plated at $3 \times 10^5$ cells/ml in 35 mm cell culture dishes. HEK 293-A cells were treated with dimethyl sulfoxide (DMSO) or ML141 (10 μM; 24 hr, 37 °C, 5% $CO_2$). The next day, cells were treated with DMSO, ML141 (10 μM), rifampin (0.01 mg/L), or cotreatment of ML141 (10 μM) and rifampin (0.01 mg/L; 1 hr, 37 °C, 5% $CO_2$). HEK 293-A cells were harvested and resuspended in FACS buffer, then stained with propidium iodide (PI). Samples were analyzed using a MACSQuant Analyzer 10 flow cytometer. Data are represented as percent $PI^+$ cells ± SEM ($P > 0.05$ by one-way ANOVA followed by Holm-Sidak's post-hoc analysis, $n = 3$/treatment). Data are pooled from two independent experiments. (C) Representative histogram from PI assay indicates similar levels of PI uptake by HEK 293-A cells treated with DMSO (green), ML141 (black), rifampin (purple), or rifampin with ML141 as cotreatment (red). The overlay of histograms reveals nearly overlapping peaks for each treatment.

## DISCUSSION

In this study, we report cotreatment of ML141 with rifampin decreases intracellular infection more than rifampin monotherapy (Fig. 2). Improved clearance is achieved through mechanisms that sustain host cell viability and host membrane integrity (Fig. 3) in the absence of detectable improvement of rifampin bactericidal activity (Fig. 4).

In contrast, cotreatment of cholesterol-lowering simvastatin with rifampin yielded no detectable improvement in bacterial clearance (Fig. 5). To explore the differential response between ML141 and simvastatin, we compared effects on host membrane permeability. We found that in contrast to ML141, simvastatin increases membrane permeability. This finding is consistent with earlier reports by our group and others that simvastatin induces membrane permeability in specific cell types (*Caffo et al., 2019*; *Chow et al., 2010*). Statins exert pleiotropic effects through inhibition of multiple intermediates within the cholesterol biosynthesis pathway, including loss of membrane integrity through decreased synthesis of the intermediate mevalonate (*Chimento et al., 2018*; *Rauthan & Pilon, 2011*). Yet, loss of membrane integrity is not a universal response to statins (*Horn et al., 2008*; *Rodrigues et al., 2009*). Thus, it is plausible that examination of simvastatin across multiple cell types, timepoints and dosages may have yielded a response similar to ML141.

We found the increase in membrane permeability by simvastatin was reversed by rifampin cotreatment. The return to baseline in response to cotreatment may be due to host-directed effects of rifampin. Rifampin not only acts through inhibition of bacterial RNA polymerase but also through host-directed responses, including induction of members of a subclass of the mammalian ATP-binding cassette family, the multidrug resistance protein (MRP) transporters (*Fromm et al., 2000*). Statin drugs can undergo efflux from the cell via these MRP transporters (*Knauer et al., 2010*). Our observation that membrane permeability returned to baseline in the cotreatment group would be consistent with statin efflux driven by rifampin induction of these transporters. Similarly, the failure of cotreatment to improve bacterial clearance also would be consistent with rifampin-driven statin efflux. However, such conclusions await further experimental evidence.

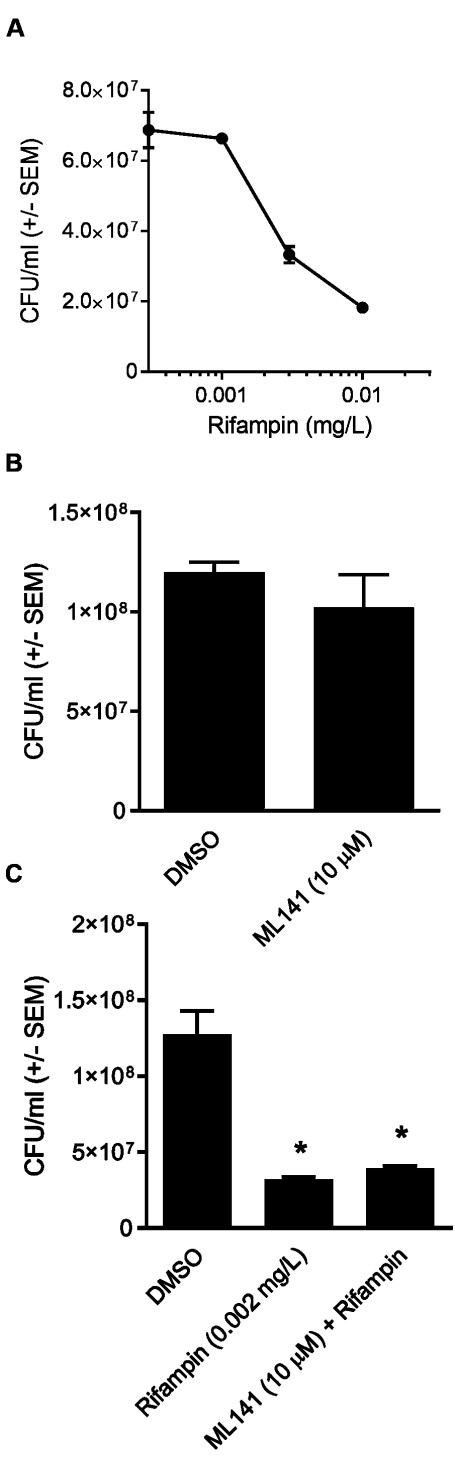

**Figure 4** **Rifampin bactericidal activity is not enhanced by ML141.** (A) *Rifampin IC50 under axenic conditions is 0.002 mg/L.* $3 \times 10^7$ *Staphylococcus aureus* colony forming units (CFU) were incubated with increasing concentrations of rifampin or dimethyl sulfoxide (DMSO; 1 hr, 37 °C, 5% $CO_2$). Serial dilutions were plated on tryptic soy agar (24 hr, 37 °C) and colony 

**Figure 4 (…continued)**
counts were performed. Data are represented as CFU/ml ± SEM ($n = 3$/treatment). (B) *ML141 bactericidal activity not detected at concentrations used.* Following treatment of $3 \times 10^7$ *Staphylococcus aureus* colony forming units (CFU) with dimethyl sulfoxide (DMSO) or ML141 (10 μM; 1 hr, 37 °C, 5% $CO_2$ ), serial dilutions were plated onto tryptic soy agar (24 hr, 37 °C) and colony counts were performed. Data are represented by CFU/ml ± SEM ($P > 0.05$ by one-way ANOVA followed by Holm-Sidak's post-hoc analysis, $n = 3$/treatment). (C) *No detectable enhancement of rifampin bactericidal activity by ML141 at concentrations tested.* Following treatment of $3 \times 10^7$ *Staphylococcus aureus* CFU with dimethyl sulfoxide (DMSO), ML141 (10 μM), rifampin (0.002 mg/L), or ML141 (10 μM) in combination with rifampin (0.002 mg/L; 1 hr, 37 °C, 5% $CO_2$), bacteria were plated and quantified as described above. Data are represented by CFU/ml ± SEM (* less than DMSO, $P < 0.05$ by one-way ANOVA followed by Holm-Sidak's post-hoc analysis, $n = 3$/treatment).

Differences in the mode-of-action of simvastatin and ML141 also may contribute to the differences observed in response to cotreatment. Simvastatin inhibits 3-hydroxy-3-methylglutaryl coenzyme A (HMG-CoA) reductase, the rate limiting enzyme in the cholesterol biosynthesis pathway (*Hennessy et al., 2016*). Inhibition of HMG-CoA reductase decreases synthesis of cholesterol, as well as isoprenoid intermediates synthesized in the pathway (*Greenwood, Steinman & Zamvil, 2006*). Isoprenoid intermediates can serve as membrane anchors for CaaX domain containing proteins, including CDC42 (*Greenwood, Steinman & Zamvil, 2006*), and decreased synthesis of isoprenoid intermediates limits CDC42 membrane localization (*Horn et al., 2008*). Although simvastatin inhibits membrane localization of CDC42, activation of CDC42 by GTP binding within the activation site is sustained (*Stankiewicz et al., 2010*). This is in contrast to ML141, an allosteric inhibitor with specificity for human CDC42 (*Hong et al., 2013*; *Surviladze et al., 2010*). ML141 dissociates GTP within the activation site of CDC42, decreasing CDC42 activation (*Hong et al., 2013*). Another distinction between simvastatin and ML141 is that simvastatin inhibits membrane localization of additional CaaX-domain containing proteins, including RAC and RHO (*Liao & Laufs, 2005*). Thus, simvastatin affects multiple small GTPases, whereas ML141 has demonstrated specificity for CDC42, with no inhibitory activity detected toward RAC or RHO (*Hong et al., 2013*). These distinctions in the mode-of-action and underlying pharmacology of simvastatin and ML141 could contribute to the observed differences in response to cotreatment.

Our findings indicate not only the promise of host-directed therapeutic approaches such as ML141, but also potential limitations of combinatorial therapies, such as the use of simvastatin with rifampin. Targeting host cell invasion may indeed have therapeutic benefit when done in combination with antibiotics, but careful examination of underlying mechanisms continues to be warranted.

## CONCLUSIONS

We sought to determine whether cotreatment of the host-directed therapeutics ML141 or simvastatin with the lipophilic antibiotic rifampin enhances clearance of intracellular *S. aureus*. We found cotreatment of ML141 with rifampin enhanced rifampin efficacy, while cotreatment of simvastatin with rifampin failed to improve rifampin efficacy. Simvastatin monotherapy increased host cell permeability, while ML141 monotherapy

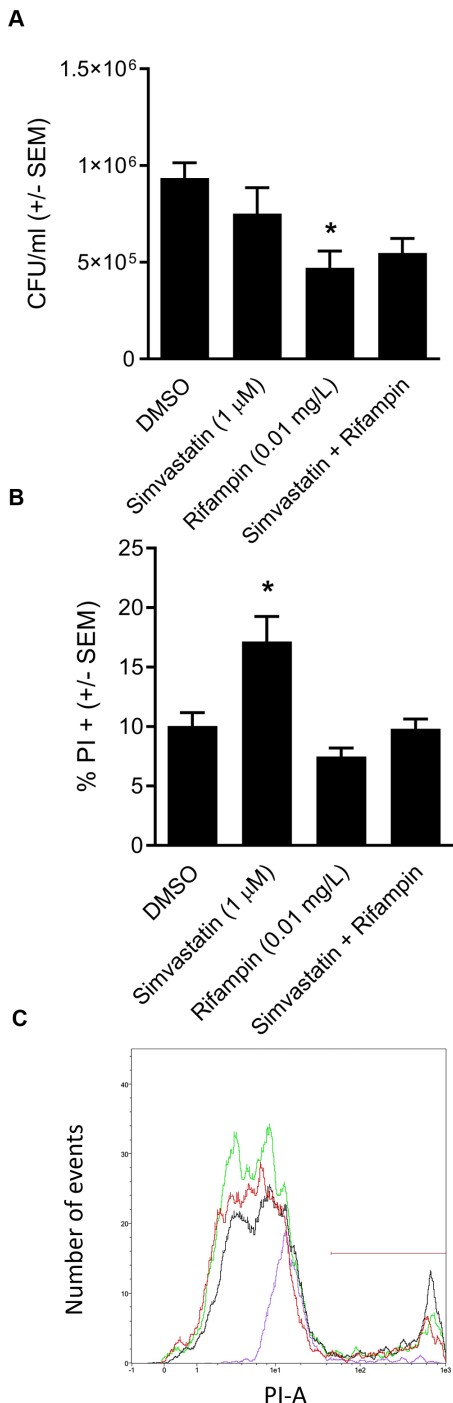

**Figure 5** **Analysis of simvastatin cotreatment with rifampin yields differential results from those of ML141.** (A) *Cotreatment of simvastatin with rifampin fails to improve rifampin efficacy.* HEK 293-A cells were plated at $3 \times 10^5$ cells/ml in 35 mm cell culture dishes. The following day, HEK 293-A cells were treated with dimethyl sulfoxide (DMSO) or simvastatin (continued on next page...)

**Figure 5 (...continued)**
(1 $\mu$M; 24 hr, 37 °C, 5% $CO_2$). The next day, HEK 293-A cells were infected with $3 \times 10^7$ *Staphylococcus aureus* colony forming units (CFU; 1 hr, 37 °C, 5% $CO_2$) then were treated with DMSO, simvastatin (1 $\mu$M), rifampin (0.01 mg/L), or cotreatment of simvastatin (1 $\mu$M) with rifampin (0.01 mg/L; 1 hr, 37 °C, 5% $CO_2$). Extracellular bacteria were killed using gentamicin (50 $\mu$g/ml) and lysostaphin (20 $\mu$g/ml; 45 min, 37 °C, 5% $CO_2$) and intracellular *S. aureus* were harvested using 1% saponin (20 min, 37 °C, 5% $CO_2$). Serial dilutions were plated onto tryptic soy agar (24 hr, 37 °C) and colony counts were performed. Data are represented by CFU/ml ± SEM (* less than DMSO, $P < 0.05$ by one-way ANOVA followed by Holm-Sidak's post-hoc analysis, $n = 3$/treatment). Data are pooled from two independent experiments. (B) *Increase in HEK 293-A cell membrane permeability by simvastatin is reversed by cotreatment.* HEK 293-A cells were plated at $3 \times 10^5$ cells/ml in 35 mm cell culture dishes. HEK 293-A cells were treated with DMSO or simvastatin (1 $\mu$M; 24 hr, 37 °C, 5% $CO_2$). The next day, cells were treated with DMSO, simvastatin (1 $\mu$M), rifampin (0.01 mg/L), or cotreatment of simvastatin (1 $\mu$M) and rifampin (0.01 mg/L; 1 hr, 37 °C, 5% $CO_2$). HEK 293-A cells were harvested and resuspended in FACS buffer, then stained with propidium iodide (PI). Samples were analyzed using a MACSQuant Analyzer 10 flow cytometer. Data are represented as percent PI$^+$ cells ± SEM (* greater than DMSO, rifampin, and simvastatin/rifampin cotreatment, $P < 0.05$ by one-way ANOVA followed by Holm-Sidak's post-hoc analysis, $n = 3$/treatment). (C) *Representative histogram demonstrates uptake of PI by simvastatin-treated HEK 293-A cells is reversed by cotreatment with rifampin.* Using MACSQuantify software, samples from PI-assay were analyzed and representative histogram generated of distribution of HEK 293-A cells treated with DMSO (green), simvastatin (black), rifampin (purple), or rifampin with simvastatin as cotreatment (red). Overlay of histograms reveals the highest peak is from simvastatin sample and that the cotreatment peak is similar to that of DMSO.

did not. Increases in host cell permeability in response to simvastatin were reversed by rifampin. Differences in the underlying pharmacology of simvastatin and ML141 may contribute to differences observed in response to cotreatment and should be considered when assessing the efficacy of use with antibiotics.

### Funding
This work was supported by the Indiana Academy of Science, the Ball State University Department of Biology, and the Ball State University Sponsored Projects Administration ASPiRE Grant Program. The funders had no role in study design, data collection and analysis, decision to publish, or preparation of the manuscript.

### Grant Disclosures
The following grant information was disclosed by the authors:
Indiana Academy of Science.
The Ball State University Department of Biology.
The Ball State University Sponsored Projects Administration ASPiRE Grant Program.

### Competing Interests
The authors declare the following patents.
1. METHODS FOR TREATING BACTERIAL INFECTION
Co-inventors: Susan McDowell, Larry Sklar, Mark Haynes, Robert Sammelson
Patent No.: US 9,259,415 B2
Date of Patent: Feb. 16, 2016

2. METHODS FOR TREATING BACTERIAL INFECTION
Co-inventors: Susan McDowell, Larry Sklar, Mark Haynes, Robert Sammelson
Patent No.: US 9,763,967 B2
Date of Patent: Sep. 19, 2017
3. METHODS FOR TREATING BACTERIAL INFECTION
Co-inventors: Susan McDowell, Larry Sklar, Mark Haynes, Robert Sammelson
Patent No.: US 2018/0021357 A1
Date of Patent: Jan. 25, 2018.

## Author Contributions

- Melissa D. Evans and Susan McDowell conceived and designed the experiments, performed the experiments, analyzed the data, prepared figures and/or tables, authored or reviewed drafts of the paper, and approved the final draft.
- Robert Sammelson conceived and designed the experiments, authored or reviewed drafts of the paper, and approved the final draft.

## Patent Disclosures

The following patent dependencies were disclosed by the authors:
  1. METHODS FOR TREATING BACTERIAL INFECTION
  Co-inventors: Susan McDowell, Larry Sklar, Mark Haynes, Robert Sammelson
  Patent No.: US 9,259,415 B2
  Date of Patent: Feb. 16, 2016
  2. METHODS FOR TREATING BACTERIAL INFECTION
  Co-inventors: Susan McDowell, Larry Sklar, Mark Haynes, Robert Sammelson
  Patent No.: US 9,763,967 B2
  Date of Patent: Sep. 19, 2017
  3. METHODS FOR TREATING BACTERIAL INFECTION
  Co-inventors: Susan McDowell, Larry Sklar, Mark Haynes, Robert Sammelson
  Patent No.: US 2018/0021357 A1
  Date of Patent: Jan. 25, 2018

## Data Availability

  The raw data are available in a Supplemental File.

## Supplemental Information

Supplemental information for this article can be found online at http://dx.doi.org/10.7717/peerj.10330#supplemental-information.

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
