# Peer review of "Differential effects of cotreatment of the antibiotic rifampin with host-directed therapeutics in reducing intracellular Staphylococcus aureus infection"

_PeerJ, doi:10.7717/peerj.10330_

## Round 0.1 · original submission · Major Revisions

Please revise the manuscript to address reviewer comments listed below.

Reviewer 1 ·

Basic reporting

In the research article titled "Differential effects of cotreatment of the antibiotic rifampin with host-directed therapeutics in reducing intracellular staphylococcus aureus infection", the authors have tried to study the impact of cotreatment of ML141/simvastatin with rifampin against S. aureus infections. Rifampin is a front line drug for treating S. aureus infections but is associated with bacterial resistance. the authors have explored a possibility of combinatorial therapy of using an anti-virulent molecule along with antibiotic rifampin to check for treating S. aureus. This concept is fast gaining attention for various infectious diseases.

Experimental design

No comment

Validity of the findings

No comment

Additional comments

The manuscript is well written with supporting evidence, however the following points needs to be addressed.

1. One of my biggest concern is missing control for all the experiments. Neither positive nor negative controls were used in the experiments. using these controls can help us to understand how efficiently the experiment worked.

2. There is inconsistency in the number of cells used for various experiments which the authors have not reasoned. proper justification is required for why for Figure 1, HEK293-A cells were infected at a MOI of 2 for figure 1A and MOI of 100 for figure 1B.
Similarly, the authors have used 3 X 105 cells/ml for all the experiments where 5 X 104 cells were used for Cell viability experiment in Figure 2A. It will be better if the authors reasons why different amount of cells were used for these experiments.

3. The results could have been explained better. Details regarding the experiments/assay performed and what observations were made were missing in the Results section.

Reviewer 2 ·

Basic reporting

1) Provide the structure of rifampin, simvastatin, and ML141 as a supplementary file.
2) Describe what is ML141 and its mechanism of action in detail.
3) Flow cytometer graphs should be provided and discussed.
4) Add literature regarding the use of these drugs against S. infections.

Experimental design

1) The experiments were performed only with one strain of S. aureus; please include a minimum 3 strains of S. aureus in the study to validate the findings. If possible, antibiotic rifampin-resistant strain should be included as a control.
2) What are the minimum inhibitory values of the rifampin, simvastatin, and ML141 against S. aureus in vitro? Please provide in vitro synergistic study data by incubating the bacteria with R, S, ML alone, and in combination against the S. aureus strains (minimum 3 strains should be included.).
3) For the co-treatment assay, what is the basis to select the following concentrations? ML141 (10 μM), simvastatin (1 μM), rifampin (0.01 mg/L), ML141 (10 μM) with rifampin (0.01 mg/L), or simvastatin (1 μM) with rifampin (0.01 mg/L) ?
4) HEK Cells should be stained with fluorescent dyes and imaged and the live dead cells images should be provided in the support of the experimental claims.
5) In Rifampin bactericidal activity assay, different concentrations of antibiotics and the ML141 should be used to confirm that the bactericidal activity is not enhanced.

Validity of the findings

1) Why simvastatin and rifampin yield differential results and rifampin-ML141 not? The authors should discuss this point in detail and provide multiple hypothesis against these results. These hypotheses should include literature citations.

·

Basic reporting

The manuscript is simple and well written.

Experimental design

Research question addressed in this manuscript is provides new knowledge on host directed therapies.

Validity of the findings

Overall the data is well represented.

Additional comments

The manuscript by Evans et al., discusses the effect of rifampin therapy in combination with host directed ML141 therapy in S. aureus infection treatment. Overall the topic is interesting and the results are useful for further research. My specific comments are:
1. In the abstract, results should be more clearly written with exact changes observed with cotreatment as compared to mono-therapy or no treatment.
2. In methods, the co-treatment method is not clear. The authors have discussed both pretreatment and cotreatment in this section, which is getting a bit confusing. Please modify.
3. The authors have performed pretreatment assays with ML141. How is it relevant in real infection cases, would the patients will keep taking ML141 as precautionary measure?
4. Results, line 163-165, “We next assessed whether the reduction in the number of intracellular bacteria was due to…” The authors need to explain the results in more detail instead of directly writing the final results, how they address this question and what experiments were performed.

---

## Round 0.2 · Minor Revisions

Thank you for revising the manuscript. Please address reviewer comment pertaining to figure 3c.

Reviewer 1 ·

Basic reporting

The authors have addressed the reviewers comments positively. The current version of the manuscript looks more convincing.

Experimental design

None

Validity of the findings

None

Reviewer 2 ·

Basic reporting

The authors have responded to all the comments.

Experimental design

The authors have responded to all the comments.

Validity of the findings

The authors have responded to all the comments.

Additional comments

The authors have responded well to the previous critiques.

·

Basic reporting

The manuscript is overall simple and provides enough background.

Experimental design

Overall experimental design is sufficient.

Validity of the findings

Conclusions are well stated and linked to manuscriot.

Additional comments

Overall I am satisfied with author response and I do not have any major concern on the manuscript. I have one minor comment.
1) In Figure 3c and %c authors should try to label X axis properly, It is unclear how authors did not use log axis beyond 104. This analysis is ok but a more clear representation will help this study further.

---

## Round 0.3 · accepted · Accept

Thank you for addressing all comments. Your manuscript has been accepted for publication.